# Addressing Critical Issues Related to Storage and Stability of the Vault Nanoparticle Expressed and Purified from *Komagataella phaffi*

**DOI:** 10.3390/ijms24044214

**Published:** 2023-02-20

**Authors:** Giulia Tomaino, Camilla Pantaleoni, Diletta Ami, Filomena Pellecchia, Annie Dutriaux, Linda Barbieri, Stefania Garbujo, Antonino Natalello, Paolo Tortora, Gianni Frascotti

**Affiliations:** 1Department of Biotechnology and Biosciences, University of Milano-Bicocca, I-20126 Milano, Italy; 2Université Paris Cité, CNRS, Institut Jacques Monod, F-75013 Paris, France

**Keywords:** vault nanoparticle purification, major vault protein, *Komagataella phaffii* expression system, transmission electron microscopy, Fourier-transform infrared spectroscopy

## Abstract

The vault nanoparticle is a eukaryotic assembly consisting of 78 copies of the 99-kDa major vault protein. They generate two cup-shaped symmetrical halves, which in vivo enclose protein and RNA molecules. Overall, this assembly is mainly involved in pro-survival and cytoprotective functions. It also holds a remarkable biotechnological potential for drug/gene delivery, thanks to its huge internal cavity and the absence of toxicity/immunogenicity. The available purification protocols are complex, partly because they use higher eukaryotes as expression systems. Here, we report a simplified procedure that combines human vault expression in the yeast *Komagataella phaffii*, as described in a recent report, and a purification process we have developed. This consists of RNase pretreatment followed by size-exclusion chromatography, which is far simpler than any other reported to date. Protein identity and purity was confirmed by SDS-PAGE, Western blot and transmission electron microscopy. We also found that the protein displayed a significant propensity to aggregate. We thus investigated this phenomenon and the related structural changes by Fourier-transform spectroscopy and dynamic light scattering, which led us to determine the most suitable storage conditions. In particular, the addition of either trehalose or Tween-20 ensured the best preservation of the protein in native, soluble form.

## 1. Introduction

Vaults are natural ribonucleoprotein nanoparticles found in several eukaryotes [1,2]. In their molecular assembly, the 99-kDa major vault protein (MVP) is present in 78 copies and generates a barrel-like, natural “nanocapsule” consisting of two symmetrical halves [3], which encloses other molecular components, i.e., the 193 kDa vault poly(ADP-ribose) polymerase, the 290 kDa telomerase-associated protein-1 (TEP1) and one or more small untranslated RNAs [4,5,6,7,8]. Overall, the molecular mass of vault particles amounts to about 13 MDa; their size is 72.5 × 41 × 41 nm, with an internal cavity volume of 5 × 10^4^ nm^3^. A well-assembled vault structure can also be produced by expressing the sole MVP, as originally shown in insect cells [9,10]. Although the physiological roles of this nanocomplex are only partially understood, numerous reports highlight its involvement in several pro-survival and cytoprotective actions [11].

Thanks to the aforementioned properties, this macromolecular assembly has attracted huge interest as a tool for drug/gene/vaccine delivery often targeted to cancer cell lines [12,13,14]. Indeed, not only can it accommodate large numbers of cargo molecules, including toxic and hydrophobic ones [15], but can also be targeted to specific cell surface receptors, provided it is bound to suitable peptides or antibodies via chemical [16] or genetical approaches. In particular, a vault variant was genetically engineered so as to carry a protein A fragment at the C-terminus. It could bind IgGs with high affinity, thus representing a general tool to target the nanoparticle to specific surface antigens [17]. Likewise, an MVP variant was produced in fusion with the peptide pVI at the C-terminus, which allows endosomal escape and improves penetration into target cells [18].

Nevertheless, the standard production protocol of vault nanoparticles is still burdensome and labor intensive due to the complexity of both the aforementioned baculovirus–insect cell expression system and the purification procedure so far available, which includes different ultracentrifugation and gradient centrifugation steps [9]. Recently, a more simplified procedure was developed by expressing a His-tagged vault variant in human embryonic kidney cells, which made it possible to affinity-purify the nanocomplex [19]. The same research group subsequently reported an *Escherichia coli*-based protocol, which unquestionably represents a substantial improvement in terms of process simplification [20]. Yet, it might suffer from obvious restraints related to the production of a non-natural, His-tagged variant, mainly in view of its employment as a nanovector to be administered to cells or whole organisms.

Thus, the most manageable expression system available to date for the production of authentic vault is based on the use of the methylotrophic yeast *Komagataella phaffii* (formerly *Pichia pastoris*) due to the ease of yeast cultivation compared with insect and mammalian cells [21]. Furthermore, we recently developed a straightforward purification procedure of the vault nanoparticle expressed in the baculovirus–insect cell system, which only consisted of a dialysis and a size-exclusion chromatography (SEC) [22]. Here, we present a novel procedure, whereby human vault expressed in *K. phaffii* was purified via a simple, SEC-based protocol, which also includes an RNase pretreatment of cell-free extracts obtained as supernatants from homogenate centrifugation. Pure preparations of authentic vault nanoparticles were thus obtained, as shown by SDS-PAGE, Western blotting and transmission electron microscopy (TEM). The ethidium bromide staining of agarose gel also demonstrated virtually complete removal of contaminating RNA.

## 2. Results

### 2.1. Production Procedure of Pure Vault Particle from K. phaffii

One major requirement in experimental designs that make use of the vault nanoparticle for either biological investigations or biotechnological applications is the availability of a manageable procedure for the production of pure preparations. We formerly developed a protocol that, starting from transfected insect cell extracts [9], led to the isolation of pure human protein through a purification procedure only consisting of a dialysis and a subsequent Sepharose CL-6B SEC [22]. Here, we present a novel protocol that takes advantage of the formerly developed yeast-based expression system [21]. This is far simpler than that based on insect cells and, no less important, should also allow a much easier scale up. We harvested the cells in the late exponential phase, when the culture attained its maximum biomass density and disrupted them, as described in Material and Methods (Section 4.4). In preliminary experiments, after cell disruption, we centrifuged the resulting homogenate at 20,000× *g* and directly subjected the supernatant (henceforth referred to as cell extract) to Sepharose CL-6B SEC, thus obtaining apparently pure preparations, as assessed in SDS-PAGE. However, we became aware of a major RNA contamination, quite likely ribosomal, as revealed by DNase-resistant ethidium bromide staining in agarose gel electrophoreses of the eluted fractions. To remove this contaminant, we adopted, with minor modifications, a previously published procedure [9], whereby we incubated the cell extract with RNases A and T1 (Section 4.4). Following this treatment, agarose gel electrophoresis actually revealed an almost complete disappearance of a broad, ethidium bromide-stained band and an equally migrating Coomassie-stained one (Figure 1A), which quite plausibly represent ribosomal contaminants. In contrast, the vault protein, which was detected in the same lanes as a slower-migrating Coomassie-stained band and identified by Western blotting, was completely unaffected by the treatment. The RNase-treated extract was then subjected to SEC and the eluted fractions again analyzed by agarose gel electrophoresis and subsequent staining according the same protocol adopted for the cell extract. This revealed the virtually complete absence of RNA contaminants in the void volume fraction, whereas residual RNA components were still visible in the second-eluted fraction (Figure 1B).

Finally, using SDS-PAGE, we checked the SEC-eluted fractions for possible protein contaminations (Figure 2). Coomassie staining revealed a major band both in the void volume fraction and in the following one, positively identified as MVP in Western blotting, and negligible amounts of protein contaminants. However, the latter fraction had to be discarded due to RNA contamination, as detected in agarose gel electrophoresis (Figure 1B). Based on these observations, in our purification protocol, we routinely identified and discarded RNA-containing fractions and recovered pure RNA-free vault in the void volume fraction, typically yielding 1.5 mg of protein per g of yeast wet weight.

### 2.2. TEM Reveals Well-Assembled Vault Nanoparticles

The TEM of specimens from the void volume fraction only displayed nanoparticles whose morphology and size were consistent with those expected, thus confirming both the homogeneity of the preparation and the occurrence of correctly assembled MVP, yielding authentic vault nanoparticles (Figure 3A1). Conversely, the second-eluted fraction revealed several particles unrelated to vault in size and morphology (Figure 3A2). In keeping with our electrophoretic analyses (Figure 1B), we assume that they are aggregated ribosomal proteins released by RNase treatment. Indeed, it is well documented that such proteins undergo aggregation, once the partner RNA components are degraded [23]. The two fractions were also subjected to dynamic light scattering (DLS), the relevant spectra being shown in Figure 3B1,B2 as number-based particle size distribution. In fact, it is suggested that, when comparing size assessments obtained by electron microscopy with DLS measurements, the number distribution is more suitable for comparisons [24]. At the same time, it is generally agreed that this parameter will provide reliable estimates provided the relevant autocorrelation profile displays a single exponential decay, diagnostic of a monodisperse preparation. As shown in Appendix A, this is the case for the samples analyzed. The analysis detected a distinctly smaller hydrodynamic radius in the second-eluted fraction compared to the first one, in keeping with the TEM images. Overall, these observations highlight the substantial role of RNase treatment in obtaining pure vault preparations.

### 2.3. Determining Optimal Conditions for Vault Storage and Stability using FTIR Spectroscopy and DLS 

When freshly eluted from the SEC column, the vault preparations did not display any detectable protein loss in the supernatants upon centrifugation at 20,000× *g* for 20 min, although tiny amounts of precipitated protein could be identified in the FTIR (Fourier-Transform Infrared) Spectroscopy analysis, as discussed below, thanks to the remarkable sensitivity of this analytical technology. In contrast, when either incubated at +4 °C or frozen and thawed under different conditions, the same preparations displayed a significant propensity to generate precipitates, suggestive of ongoing aggregation. We thus set out to explore the vault’s stability in solution under different storage conditions, with the aim of both identifying those compatible with the best recovery of soluble, native protein and providing insight into the structural changes responsible for its aggregation (Table 1). When incubated at +4 °C, a loss of approximately two-thirds of the protein in the supernatant was observed after only one week. Thus, this condition was proven to be unsuitable for long-term storage.

Upon incubation under frozen conditions, a better preservation of soluble protein was generally achieved. In particular, the frozen vault was subjected to short- and long-term storage (1 day and 7 weeks, respectively), under different conditions, including the addition of either trehalose or the non-ionic surfactant Tween-20. In the presence of trehalose, we also assessed the recovery of soluble protein after sample lyophilization and reconstitution (Table 1). In fact, this sugar has been employed as an additive in vault lyophilization [17]. Likewise, the capability of Tween-20 to prevent protein aggregation and improve thermodynamic stability is well documented [25,26].

The different freezing–thawing treatments also resulted in a significant decrease in soluble protein, in the range 35–55%, as assessed after one-day storage. A further, smaller decline in soluble protein was detected in the frozen samples at the longest storage time, i.e., 7 weeks, with the best recoveries being obtained in the presence of Tween-20. These observations make it apparent that protein loss mainly takes place during the freezing process rather than during the subsequent storage.

To further structurally characterize the vault nanoparticle after incubation under the above-described conditions, as well as to provide insight into possible structural changes related to protein precipitation, both supernatants and pellets were investigated by FTIR in attenuated total reflection (ATR) mode (Figure 4). In particular, we analyzed the Amide I band in the second derivatives of the protein IR absorption spectra. This band is due to the C=O stretching vibration of the peptide bond and is sensitive to the protein secondary structures as well as to the formation of intermolecular β-sheets in protein aggregates [27,28].

The second derivative spectrum of the freshly purified vault (Figure 4A) displayed a main peak at ~1655 cm^−1^ that can be assigned to α-helical and random coil structures. Less intense peaks were observed at ~1638 cm^−1^, due to the native β-sheets, and at ~1688 cm^−1^. This latter broad band can be assigned to β-sheet and β-turn structures. The relative intensities of the ~1655 cm^−1^ and ~1638 cm^−1^ Amide I components are well in agreement with the structural data currently available for the native protein [29], indicating an overall higher content of α-helix and random coil structures (~51%) compared with β-sheets (~28%). The spectrum of the supernatant was almost superimposable to that of the total sample (Appendix A), whereas the pellet spectrum displayed a slightly reduced intensity of the Amide I components assigned to the native protein structures and a new component at ~1629 cm^−1^, which appears as a very low intensity shoulder and can be assigned to the formation of intermolecular β-sheet structures typical of protein aggregates. Interestingly, the strong intensity of the peaks associated with the native structures in the pellet suggests that, rather unusually, a substantial fraction of native conformation is present in the aggregated protein.

The ATR-FTIR analysis was also performed on vault preparations subjected to storage either at + 4 °C or after freezing under different conditions. The spectra of the supernatants were essentially indistinguishable from one another as well as from that of fresh protein and were representative of a protein in the native state (Figure 4B–F).

However, for the sake of completeness, it should also be noted that the slight spectral differences observed in the supernatant samples in the presence of trehalose compared with the others (Figure 4E) may well be accounted for by an instrumental constraint. Specifically, they essentially result from the so-called z-dilution, namely, protein dilution along the z-axis due to the presence of the sugar, with resulting reduced protein accumulation in the immediate vicinity of the ATR interface [30]. Likewise, the supernatant spectrum collected in the presence of Tween-20 displayed a slightly reduced ratio of β-sheet to α-helix peak intensities (Figure 4F). Although we have no obvious interpretation for this result, the spectrum collected confirms that under these conditions, the protein also retains a native conformation. We speculate that this result may reflect the dynamic structure of the protein, as documented by its capability of adopting at least two different conformations [29]. Nevertheless, the protein preparations incubated in the presence of Tween-20 as well as of trehalose were also subjected to TEM analysis after spinning down at 20,000× *g* possible aggregates, which made it possible to detect the typical vault morphology under both conditions (Appendix A).

Furthermore, as already observed in the case of fresh protein, we found that under all employed storage conditions, the pelleted protein also retained a large fraction of native conformation, as documented by the relevant spectra that display the Amide I components observed in the respective supernatants (Figure 4B–F). Concurrently, in such samples, the ~1629 cm^−1^ component, which is representative of intermolecular β-sheets, only appeared as a shoulder and not as a well-resolved peak, albeit more intense than that detected in the pellet of the fresh protein (Figure 4A). These findings clearly show that a native-like structure must be largely prevailing in the pellet, wherein the vault molecules must otherwise interact with one another in some way to give rise to insoluble matter. Thus, intermolecular β-sheets, which are a typical hallmark of denaturation-induced aggregation both in vitro and in vivo [31], are scarcely represented in the pellets. Currently, we do not have any obvious interpretation for this phenomenon. However, we deem it worth mentioning that even TEM images of soluble preparations are frequently suggestive of vaults’ propensity to generate clusters with no ordered aggregation pattern, wherein the individual nanoparticles otherwise retain their typical conformation [22]. On the other hand, the dynamic nature of the vault nanoparticle is well documented, as mentioned above, and also results in several modes of interaction, quite likely of physiological relevance such as, for instance, the in vivo formation of intracellular tube-like structures, whereby MVP molecules interact with one another via their coiled-coil domains [32], MVP subunit exchange [33] and vault opening [34].

Finally, we also subjected our preparations to DLS analysis in order to assess their homogeneity, focusing on the frozen, long-term incubated samples, namely, the ones that are more likely to be employed for any predictable application. In line with the criteria outlined when discussing Figure 3, in Figure 5, the number-based particle size distribution is presented, and the respective correlograms and intensity plots are shown in Appendix A. The correlograms generally fitted with a single exponential decay, with the exception of the sample stored at −20 °C, the only one displaying a somewhat larger size and some degree of polydispersity, as documented by the relevant PDI. Except for this condition, which was therefore judged unsatisfactory for long-term storage, we determined nanoparticle dimensions reasonably in agreement with those determined in previous reports [19,35].

Unfortunately, inconsistent DLS measurements were obtained when analyzing vault samples incubated in the presence of 0.05% Tween-20, which prevented us from collecting reliable data under these conditions. This effect is plausibly justified by the well-known propensity of non-ionic detergents to generate a heterogeneous micelle population, wherein different numbers of protein molecules may be recruited into different micelles [36]. However, as already mentioned above, TEM analysis of the vault incubated in the presence of the detergent did not reveal any gross morphological alteration (Appendix A).

Overall, our investigations highlight a critical issue related to vault purification and handling, i.e., its propensity to aggregate, but also provide the most suitable approaches to cope with. The implications of these results are analyzed in the Discussion section.

## 3. Discussion

In the last decades, the vault nanoparticle has attracted considerable interest for both its multifaceted and as-yet only partially elucidated biological roles, and its potential as a nanovector for drug/gene delivery [2,37]. Nevertheless, most protocols available for vault production must still cope with major constraints regarding both the expression systems and the purification procedures, which are lengthy and multistep in nature. In particular, the baculovirus–insect cell system is complex and time consuming [9], whereas the human-cell-based procedure is definitely simpler [19], but both of them hardly lend themselves to a significant scale up. Conversely, the recently developed *E. coli*-based protocol actually allows a substantial simplification of the process, but only enables the production of a vault variant consisting of His-tagged MVP subunits [20]. The present contribution reports a novel protocol that overcomes these limitations by combining the use of an engineered *K. phaffii* strain as the expression system of an authentic vault [21] with a purification procedure that represents a modification of a previously reported one [22], now essentially consisting of the RNase treatment of yeast-cell-free extracts and a subsequent SEC. This avoids the need for multiple cycles of ultracentrifugation [9], as well as for a dialysis step [22]. In this way, we were able to produce about 1.5 mg of pure vault per g of yeast wet weight, which corresponds to about 50–60 mg/L culture. This yield is significantly better than that previously reported using the same host organism [21]. Other published protocols based on insect [9] or human cells [19] are not likely to perform better given the inherent limitations related to the production of the starting cellular material, regardless of the complexity of the purification procedure.

It should be also pointed out that the vault nanoparticle could be expressed constitutively thanks to its lack of toxicity in yeast cells. This made it possible to grow cells in the absence of inducers, which are generally very expensive, making the process hardly scalable.

Based on different analytical methods, we demonstrated that our vault preparations are essentially pure and, no less important, correctly folded so as to display the expected morphology, as substantiated by TEM images and DLS measurements. Thus, they are apparently of suitable quality for virtually all of the potential applications they may be destined for.

However, a critical issue became apparent during the present experimentation, namely, a significant propensity of vault preparations to generate insoluble aggregates. They appeared on incubation at 4 °C, as well as after freezing–thawing under different conditions. In addition, under all assayed conditions, they displayed a typical FTIR peak at ~1629 cm^−1^, which is representative of intermolecular β-sheets. This is a signature of protein aggregation, a process envisaging a progressive recruitment of the involved proteins via the initial formation of small oligomers, which subsequently evolve into large aggregates [31]. This is paralleled by a time-dependent increase in intermolecular β-sheets and accordingly of the relevant peak at ~1629 cm^−1^. In contrast, we surprisingly observed that the pellets collected after a one-day and a one-week incubation at 4 °C displayed virtually indistinguishable spectra (Figure 4B), which does not fit with a pattern entailing progressive structural changes, as expected during protein aggregation. Rather, it points to a two-state conversion from a soluble to an aggregated form, the latter being richer in intermolecular β-sheets. This phenomenon deserves further investigations in order to provide insight into the mode of vault association and its possible physiological relevance.

Besides the aforementioned investigations, which led to a partial characterization of the structural features distinctive of the aggregates, we set out to select the most suitable storage conditions to prevent their appearance, with the obvious intent of ensuring the best recovery of soluble, native vault during long-term storage. As shown in Table 1, it is apparent that, on the one hand, most (but not all) protein loss resulted from the freezing process; on the other hand, significant differences in soluble vault recovery were detected depending on the adopted conditions, although none of them ensured full recovery of soluble protein. In merely descriptive terms, it is apparent that storage at −80 °C, rather than at 4 °C or −20 °C, is better capable of preserving the soluble form along with its native size and secondary structure content. Moreover, trehalose addition improved, to some extent, the recovery of soluble protein, in keeping with a previous report [17]. It is worthy of note that our results also confirm that, in the presence of the sugar, vault preparations can withstand long-term storage in lyophilized form. Nevertheless, vault solubility was best preserved when stored at −80 °C in the presence of 0.05% Tween-20, a non-ionic detergent that, in our preliminary experiments, was selected among others due to its better capability to prevent aggregation. As already mentioned [25,26], its stabilizing and solubilizing effect on proteins is well known. It also did not detectably perturb the vault’s overall morphology, as apparent in the TEM images (Appendix A). Based on these results, we conclude that its stability and solubility can be better preserved when stored in the presence of either trehalose or Tween-20. However, in view of possible vault administration to cell cultures or tissues, caution should be taken when using this latter compound due to the well-known cell lytic effects exerted by detergents, even non-ionic ones [38].

To the best of our knowledge, no stability data under the conditions described in Table 1 were reported prior to the present contribution. However, we assume that our purification procedure should ensure a very good preservation of protein stability thanks to its simplicity, particularly compared with others encompassing several centrifugation steps. This is also attested by the fact that the vault nanoparticle did not undergo any appreciable aggregation when freshly eluted from the SEC column. Thus, the relative instability subsequently detected (Table 1) does not seem to be related to the previous handling.

In conclusion, the achievements of the present investigation not only deliver the most practical purification procedure to date of authentic vault, but also provide further hints regarding its structure dynamics, which might have both theoretical and practical implications.

## 4. Materials and Methods

### 4.1. Strain and Growth Media

For the production of the MVP protein, the vacuolar aspartyl protease PEP4-deficient *K. phaffii* SMD1168 (his4, ura3, pep4::URA3) strain was used [39]. The strain is available from Invitrogen (Carlsbad, CA, USA). *K. phaffii* was grown in a flask in BMDY medium (10 g/L yeast extract, 20 g/L bacto peptone, 20 g/L dextrose, 100 mM potassium phosphate buffer, pH 5.8, 13.4 g/L yeast nitrogen base without amino acids, 0.4 mg/L biotin) [40,41]. All media were from Biolife; monobasic potassium phosphate, yeast nitrogen base without amino acids and biotin were from Sigma-Aldrich. Standard liquid and plate growth were performed on YPD medium (10 g/L yeast extract, 20 g/L bacto peptone, 20 g/L dextrose, 20 g/L agar omitted in liquid medium) with or without 100 μg/mL zeocin. After electroporation, YPDS plates containing 1 M sorbitol and 100 µg/mL Zeocin were used for the selection of the transformants.

### 4.2. MVP Gene Cloning

The cDNA sequence coding for human major vault protein (MVP, GenBank accession no. BC015623.2) was PCR-amplified using the forward primer JB183-AATTAGAATTCACCATGGCAACTGAAGAGTTCATC (Eurofins), the reverse primer JB184-ATTAGGTACCTTAGCGCAGTACAGGCACCACG (Eurofins) and the pVL1393-MVP vector [22] as the template. The PCR product was then digested with EcoRI and KpnI and subcloned downstream of the glyceraldehyde 3-phosphate dehydrogenase promoter (pGAP) in the EcoRI/KpNI digested pGAPZB yeast vector DNA (Invitrogen, Carlsbad, CA) to form pGAPZB-MVP. The construct was confirmed by Sanger sequence analysis carried out by Eurofins.

### 4.3. Transformation and Selection of Positive Clones

The plasmid pGAPZB-MVP was linearized with BspHI and transformed into *K. phaffii* SMD1168 protease-deficient strain, using a Gene Pulser II electroporator (Bio-Rad Laboratories, Hercules, CA, USA), following the procedure previously described [21], with minor modifications. Briefly, electroporation was performed at 1.5 kV, 400 Ω and 25 µF by a single pulse. The transformation mixture was plated on YPDS selective agar plates. To make sure that the pGAPZ B-MVP plasmid had correctly integrated within the yeast genome, a colony PCR was performed on the Zeocin-resistant clones. A cell suspension was obtained by resuspending a small number of cells from the colonies in 10 µL of RNAse-free water. The enzyme Wonder Taq polymerase (Euroclone) was used for the amplification. The correct integration of the plasmid was also confirmed by agarose electrophoresis (2% gel). The cells were collected, lysed and subjected to sodium dodecyl sulfate–polyacrylamide gel electrophoresis (SDS-PAGE), followed by Coomassie staining and Western blot analysis. The most productive clones were selected, stored at −80 °C as glycerol stock and used in all subsequent experiments.

### 4.4. Cell Growth and Collection, and Cell-Free Extract Production

The best-producing clone was streaked on YPD plates with 100 µg/mL Zeocin and incubated at 30 °C for two days. A 1 mL suspension in sterile water of cells grown on plates was used to inoculate at OD_600_ 0.4 a 1000 mL Erlenmeyer flask containing 200 mL of BMDY medium. The flask was incubated at 30 °C under stirring at 160 rpm. After approximately 24–30 h, 100 mL aliquots of cell cultures were collected and washed with cold Milli-Q water through repeated sedimentation cycles for 5 min at 5000× *g* in a bench-top centrifuge (Eppendorf 5430R) in pre-weighted 50 mL test tubes (Falcon). The pellets were weighed, wet weight recorded and stored at −80 °C. Then, they were resuspended in breaking buffer (3 mL/g wet weight; 50 mM sodium phosphate buffer, pH 7.4, 5% glycerol (*v*/*v*), 1 mM EDTA, 1 mM dithiothreitol, 1 mM PMSF, Pierce™ Protease Inhibitor Mini Tablets EDTA-free according to the producer’s recommendations). Two-milliliter aliquots of cell suspensions were transferred into 10 mL round bottom tubes containing 1 mL of glass beads (425–600 μm diameter). The cells were disrupted by vigorous vortexing for ten periods of 1 min each, with idle intervals of 1 min in ice. The homogenate was withdrawn and centrifuged for 20 min at 20,000× *g*, 4 °C (Eppendorf 5430R). The cell-free extract was then incubated with RNase A and RNase T1 (0.5 mg and 50 U per mL of extract, respectively) for 20 min at room temperature under stirring. Then, it was centrifuged for 20 min at 20,000× *g* and 4 °C to eliminate insoluble ribosomal protein [9]. Finally, the supernatant was collected to be destined for vault purification.

### 4.5. Vault Purification

RNase-treated cell-free extract (approximately 70 mg protein content in 6 mL) was loaded onto a Sepharose CL-6B column (Healthcare Life Sciences; fractionation range 10 kDa–4 MDa; 206 mL bed volume, 42 cm height) pre-equilibrated with 20 mM Hepes, pH 7.4, 75 mM NaCl, 0.5 mM MgCl_2_. Elution was performed at 4 °C using the same buffer at a flow rate of 2 mL/min and 6 mL fractions were collected. Both in extract and in the individual fractions, protein was assayed by the bicinchoninic acid assay using the KIT QPRO-BCA (Cyanagen) and bovine serum albumin as the calibration standard.

### 4.6. Electrophoretic Analyses

The purified vault samples were subjected to SDS-PAGE (8% gel). Typically, 6–12 μg samples were applied. Gels were stained by Coomassie brilliant blue R-250 following a standard protocol. MVP identity was confirmed by Western blot analysis on PVDF membrane Immobilon (Millipore) using a primary anti-MVP antibody (rabbit monoclonal; 1:10000 dilution in PBS-5% skim milk; Abcam) and a fluorescent secondary anti-rabbit IR-800 antibody (1:16000 dilution in PBS-5% skim milk; LI-COR Biosciences). The immunoreactive signal was revealed using an Odyssey Fc instrument (LI-COR Biosciences). For nucleic acid detection, the samples (typically 3 to 10 μg protein) were run in 0.7% agarose and TAE buffer (40 mM Tris base, 20 mM acetic acid, 1 mM EDTA, pH 8.3–8.5) and ethidium bromide-stained. Staining was revealed using a Uvidoc HD6 transilluminator (Uvitec).

### 4.7. TEM

TEM images of vault particles were obtained using a Tecnai12 (RH42B) microscope (accelerating voltage: 120 kV—filament: LaB6). Samples were deposited on carbon-coated copper grids, 400 mesh, after plasma activation for 20 sec, by floating the grid onto the protein drop (20 μL, 0.5 mg/mL) for 1–2 min. The grid was then dried from liquid excess by filter paper and put on a drop of uranyl acetate (1% in PBS, pH 5.0) for 1–2 min, depending on sample concentration. Finally, the grids were dried with Whatman filter paper and analyzed.

### 4.8. FTIR Spectroscopy

The FTIR measurements were performed in ATR mode using 2 µL of the protein preparations, which were deposited on the diamond plate of the single reflection ATR device (Quest, Specac, Swedesboro, NJ, USA). Spectra were recorded after solvent evaporation as previously described [27]. The Varian 670-IR spectrometer (Varian Australia Pty Ltd., Mulgrave VIC, AU) was employed under the following conditions: 1024 scan coadditions, 2 cm^−1^ spectral resolution, 25 kHz scan speed, triangular apodization, and nitrogen-cooled Mercury Cadmium Telluride (MCT) detector. The absorption spectra, after the subtraction of the buffer spectra and, when necessary, after water vapor correction, were normalized at the Amide I band area and were smoothed using the Savitsky–Golay method (25 points) before the second-derivative calculation (Resolutions-Pro software, Varian Australia Pty Ltd., Mulgrave, VIC, Australia).

### 4.9. DLS

DLS measurements were performed by a Zeta Sizer Nano Instrument (Malvern Instruments Ltd., Amesbury, UK) operating at 4 mW of a HeeNe 633 nm laser, using a scattering angle of 90°. A disposable cuvette with 1 cm optical path length was used for the measurements. Each sample was allowed to equilibrate for 2 min prior to measurement. Three independent measurements of 60 s duration were performed at 25 °C. Calculations of the hydrodynamic diameter were performed using the Mie theory, considering the absolute viscosity and the refractive index of the material set to 1.450, Abs 0.001. The number-based hydrodynamic diameter and the autocorrelation function were determined, the latter being diagnostic of sample homogeneity, insofar as a single exponential decay profile is detected.

## Figures and Tables

**Figure 1 ijms-24-04214-f001:**
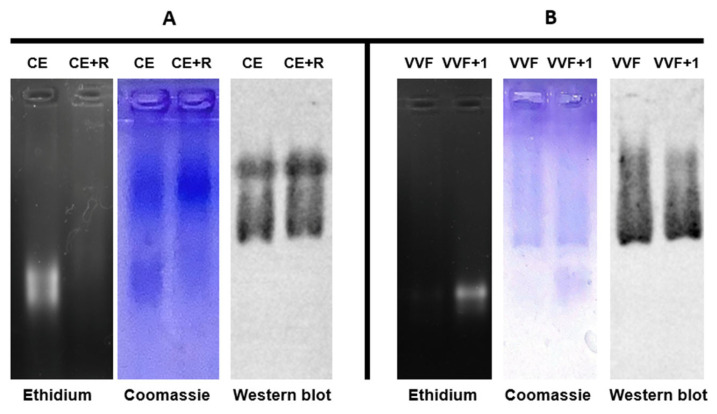
Electrophoretic characterization of a cell-free extract from the vault-expressing *K. phaffii* strain (**A**) and the purified vault (**B**). Agarose gel electrophoresis (0.7% gel) was performed on the extract before (CE) and after (CE + R) RNase treatment. Then, the gel was subjected to ethidium bromide and Coomassie staining, and Western blotting. Void volume fraction (VVF) and the following one (VVF + 1) from SEC of the RNase-treated extract was subjected to the same analytical procedures. Other details are reported in Materials and Methods (Section 4.6).

**Figure 2 ijms-24-04214-f002:**
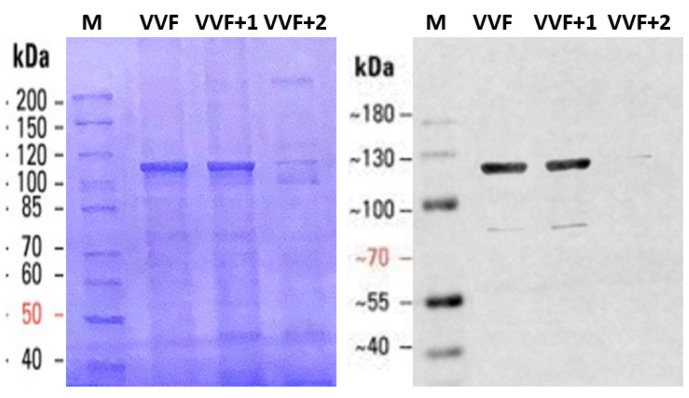
SDS-PAGE (8% gel) of vault eluted from the SEC column. The void volume fraction (VVF) and the following ones (VVF+1 and VVF+2) were subjected to electrophoresis, Coomassie stained (left panel) and Western blotted (right panel). M: standard proteins with the respective molecular weights (kDa). Other details are reported in Materials and Methods (Section 4.6).

**Figure 3 ijms-24-04214-f003:**
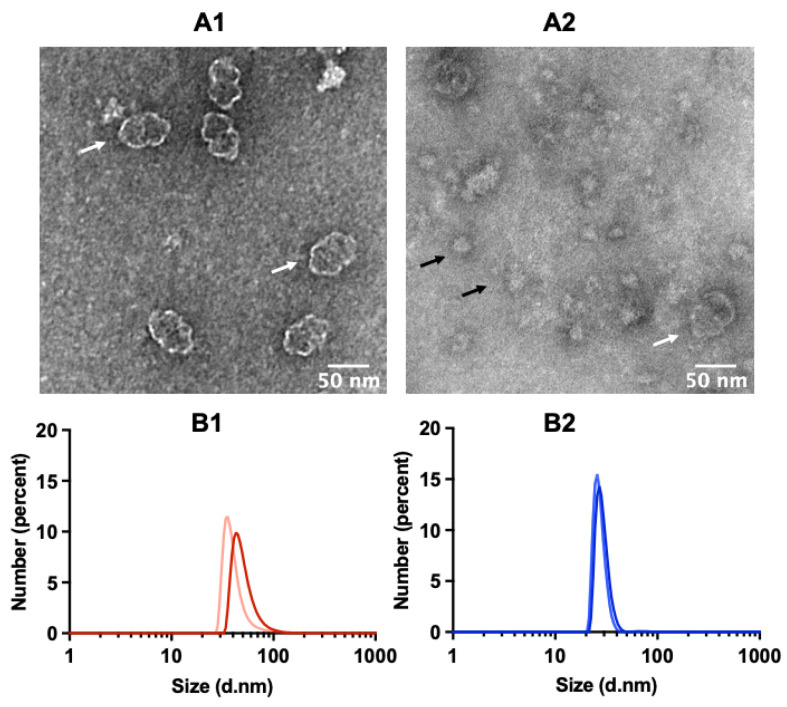
TEM images of vault nanoparticles stained by uranyl acetate, as detected in the void volume fraction (**A1**) and in the following one (**A2**). The fractions were analyzed after centrifuging at 20,000× *g* and discarding the pellet. Vault particles are indicated by white arrows; contaminating matter, quite likely ribosomal components, are indicated by black arrows. DLS of the void volume fraction and the following one are shown in (**B1**) and (**B2**), respectively. Results are presented as number-weighted particle size distributions. (**B1**): size 51.3 ± 15.6 (mean ± std. dev.); polydispersity index (PDI): 0.203. (**B2**): size 28.9 ± 4.0; PDI 0.263. Result quality: good. In each panel, two replicates of the measurement are shown. The relevant intensity and autocorrelation plots are presented in Appendix A.

**Figure 4 ijms-24-04214-f004:**
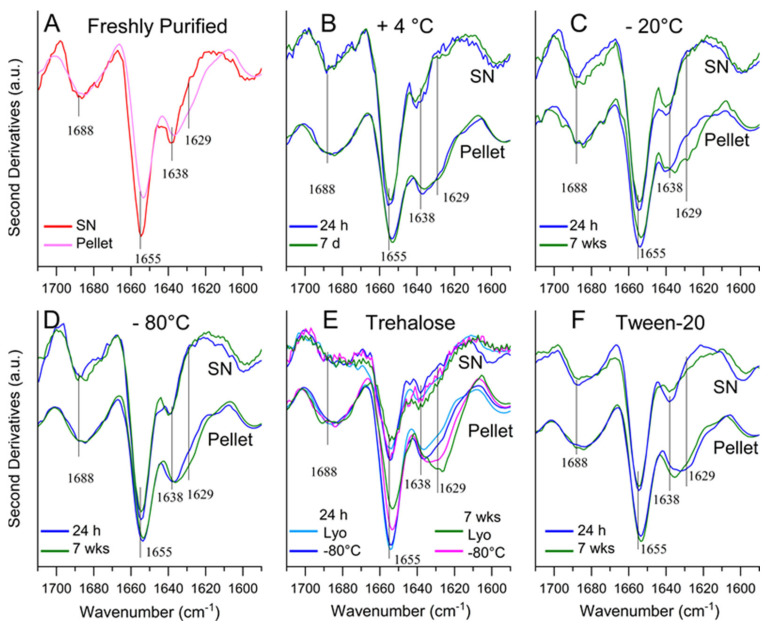
FTIR analysis of vault protein preparations. Samples were stored for the indicated times and under the indicated conditions, as shown in the individual panels. Then, second derivatives of the IR absorption spectra of supernatants (SN) and pellets after centrifugation at 20,000× *g* were analyzed. (**A**) Sample freshly eluted from the SEC column; (**B**) sample incubated at 4 °C; (**C**–**F**) samples frozen and thawed, except for panel (**E**), wherein the sample was frozen at −80 °C in 10 mg/mL trehalose and either thawed or lyophilized and reconstituted (Lyo) before the analysis; (**F**) sample frozen at −80 °C in 0.05% Tween-20. The main peak positions are reported. Spectra of total samples (analyzed prior to centrifugation) are shown in Appendix A.

**Figure 5 ijms-24-04214-f005:**
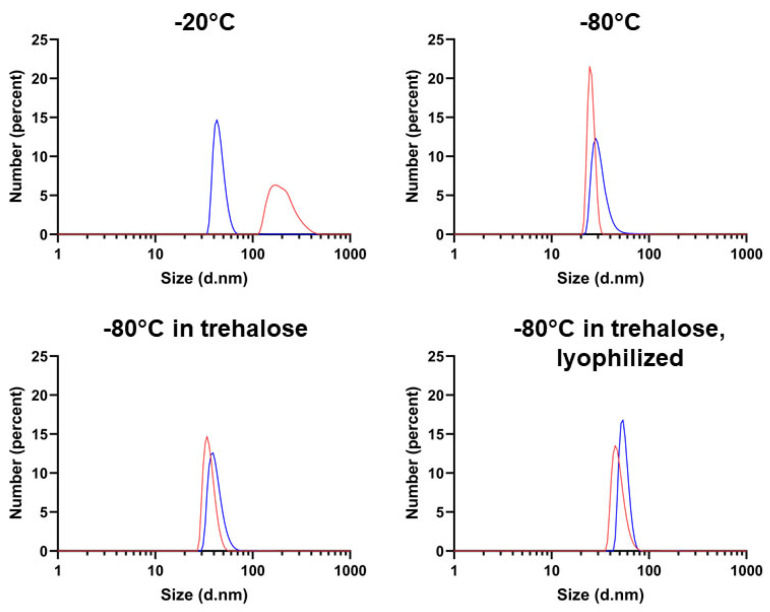
DLS of the 20,000× *g* supernatants of vault samples incubated for 7 days and otherwise as indicated in the respective panels. Results are presented as number-weighted particle size distribution. −20 °C: size 204.8 ± 17.9 (mean ± std. dev.) nm, PDI: 0.332; −80 °C: size 94.00 ± 9.09 nm, PDI 0.007; −80 °C plus trehalose: size 120.30 ± 11.89, PDI 0.044; −80 °C plus trehalose, lyophilized and reconstituted: size 148.20 ± 19.24, PDI 0.145. Result quality: good. In each panel, two replicates of the measurement are shown. The relevant intensity and autocorrelation plots are presented in Appendix A.

**Table 1 ijms-24-04214-t001:** Residual protein content in supernatants of vault preparations stored under different conditions.

Storage Conditions ^1^	1 Day	1 Week	7 Weeks
+4 °C	51 ± 4	35 ± 2	
−20 °C	44 ± 2		31 ± 7
−80 °C	52 ± 5		43 ± 6
−80 °C in 10 mg/mL trehalose	56 ± 6		50 ± 3
−80 °C in 10 mg/mL trehalose, lyophilized and reconstituted	50 ± 8		45 ± 5
−80 °C in 0.05% Tween 20	64 ± 4		59 ± 5

^1^ After storage for the indicated times, frozen samples were thawed and centrifuged at 20,000× *g* for 20 min. Then, residual protein in the supernatants was determined using the bicinchoninic acid (BCA) assay and expressed as percentage relative to zero time. Figures are mean values ± standard deviation (*n* ≥ 3).

## Data Availability

Not applicable.

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
