# Peer review of "Addressing Critical Issues Related to Storage and Stability of the Vault Nanoparticle Expressed and Purified from Komagataella phaffi"

_ijms, 2023, doi:10.3390/ijms24044214_

Round 1

Reviewer 1 Report

The manuscript presents a modified purification procedure for vault particles. The purification effect, storage and stability of the procedure is fully analyzed. It can be published after minor revision.

---the exponential coordinate for particle size distribution has poor differentiation, which can be modified.

---it seems the disparity of two replicates for two particle size distribution in Fig 3, Fig 5 are too large, which should be properly explained.  

---for the backgrounds, relevant Refs of ACS Nano 2013, 7, 889; Small 2011, 7, 1432 can be covered.

---all the Refs should be confirmed, e.g., document number for ref2, publication year for Ref 15, and etc.

Author Response

---the exponential coordinate for particle size distribution has poor differentiation, which can be modified.

We have changed accordingly Figs 3 & 5

---it seems the disparity of two replicates for two particle size distribution in Fig 3, Fig 5 are too large, which should be properly explained.

The discrepancy between the replicates is intrinsic of the DLS measurement, especially when normalized for the number-distribution and indeed the values reported in the main text do refer only to the averaged signals. On the other hand, the number distribution is more suitable for comparisons when comparing size assessments obtained by electron microscopy with DLS measurements (see Ref. 24). Anyway, we purposely also reported the raw data of DLS, in terms of intensity and correlograms in the supplementary materials. There the results show a much less pronounced divergence between the duplicates. Furthermore, in the figures, we chose to report the individual traces to fully portray the size distribution of the particles in solution and, to specifically highlight the aggregation taking place when the sample is not properly handled, as in the case of storage at -20 °C (Fig.5A). In the specific case of Fig.5A, the discrepancy between the replicates is to be associated with the less pronounced amount of correctly assembled vault. So in the revised version we explicitly state that this condition is unsuitable for long-term storage (line 274).

---for the backgrounds, relevant Refs of ACS Nano 2013, 7, 889; Small 2011, 7, 1432 can be covered.

REPLY: We have added these two quotations (lines 45-46).

---all the Refs should be confirmed, e.g., document number for ref2, publication year for Ref 15, and etc.

REPLY: We have inspected the references. Actually, there was a mistake in ref. 2 (page 70 instead of 707), line 519, but Ref 15 is correct. Thank you for reporting the mistake.

Reviewer 2 Report

The authors reported a simple and effective procedure for vault nanoparticle purification, storage and stability. This manuscript is well written and sound interesting to the field. I am still not convinced that purification using ultra-high-speed centrifuges are better than dialysis techniques. Can the authors elaborated on this and make it clear in the manuscript? Overall, I recommend publication of this work.

Author Response

The authors reported a simple and effective procedure for vault nanoparticle purification, storage and stability. This manuscript is well written and sound interesting to the field. I am still not convinced that purification using ultra-high-speed centrifuges are better than dialysis techniques. Can the authors elaborated on this and make it clear in the manuscript? Overall, I recommend publication of this work.

REPLY: we are not sure we have fully grasped this remark, as our protocol skips both ultracentrifugation and dialysis. Anyway, we have made this point clearer in the discussion (lines 310-312).

Reviewer 3 Report

Referee Report

Manuscript number: ijms-2147797

Title: Addressing critical issues related to vault nanoparticle purification, storage and stability

By Tomaino et al

Submitted to IJMS

Comments:

This manuscript reported a simplified purification protocol and process for the vault nanoparticle based on RNase pretreatment. The authors also discussed and compared the storage of the particles under different temperatures and conditions. This work is quite topical and timely focusing on the complicated purification and storage process regarding the vault protein. I only have some minor comments for further improvement:

1.       Title: The title is very general without addressing what method/protocol/process the authors proposed in this study. They may want to add more information to the title.

2.       Abstract: Some sentences are very long. For example “Here, we report … reported to date”. It is good to shorten them.

3.       Abstract: There should be a statement to mention how vault nanoparticle is applied in biomedicine (e.g. drug and probe delivery).

4.       Introduction, L42-46: Please mention some biomedical applications and examples related to the vault nanoparticle. A schematic diagram showing the features of the particle is suggested if possible.

5.       Introduction, L47-69: The authors reviewed different purification procedures and protocols. Can they compared them quantitatively so that the readers would know how good is the most updated protocol developed by the authors?

6.       Introduction: L70: The authors may want to explain how their purification process will impact the stability of the vault nanoparticle based on various storage conditions.

7.       Results: Sec. 2.1: The title should use the same style of fonts. Please see “K. phaffii”.

8.       Figure 1: Is it possible to add the scale, for example “kDa” in the vertical axis?

9.       Figure 3: Is it possible to move the “A1” and “A2” label out of the TEM image in order to make the presentation style consistent? Moreover, I cannot see Figure S1 in the manuscript.

1.   Table 1: Data are missing in the second column for the 2nd to 5th row, though I understand the reason why the data of 7 weeks is missing for the 1st row.

1.   Figure 4: I cannot see Figure S2 in the caption of Figure 4 in the manuscript.

1.   Materials and Methods, L433: “4. Transmission …” should read “4.7. Transmission …”.

Author Response

-- Title: The title is very general without addressing what method/protocol/process the authors proposed in this study. They may want to add more information to the title.

Actually, this remark has led us to reconsider the title, so we now realize it should be more informative. We have changed it by mentioning the engineered microorganism we have used to produce the nanoparticle.

-- Abstract: Some sentences are very long. For example “Here, we report … reported to date”. It is good to shorten them.

Yes, we have split the sentence into two (line 20).

-- Abstract: There should be a statement to mention how vault nanoparticle is applied in biomedicine (e.g. drug and probe delivery).

When writing an abstract, one must struggle to squeeze essential information in the allowed space. Thus, let me very respectfully point out that we have already mentioned these topics, albeit in few words (“It also holds a remarkable biotechnological potential for drug/gene delivery, thanks to its huge internal cavity and the absence of toxicity/immunogenicity”; lines 16-17).

-- Introduction, L42-46: Please mention some biomedical applications and examples related to the vault nanoparticle. A schematic diagram showing the features of the particle is suggested if possible.

We have added some more information of this kind (lines 45-46 & 48-52).However, we feel an illustrative diagram would be superfluous in this context. Many similar diagrams are already reported in different Review articles, including ours (Ref. 2).

-- Introduction, L47-69: The authors reviewed different purification procedures and protocols. Can they compared them quantitatively so that the readers would know how good is the most updated protocol developed by the authors?

In ref. 21, it is reported that vault yield from P. pastoris is 7-11 mg /L culture and that is comparable with the one obtained from Sf9 insect cell cultures. In our procedure, we produced 50-60 mg pure vault/L culture, which also corresponds to about 1.5 mg/g yeast wet weight (this latter result being already mentioned in the first submission). Thus, the previous yeast- and insect cell-based protocols attain yields significantly lower than ours.

As regards the protocol devised to purify recombinant His-tagged vault from human embryonic kidney (HEK) cell lines (ref. 19), we were unable to extract information specifying the amount of pure protein obtained from a given starting amount of cells. Nevertheless, given the type of procedure we feel it might hardly yield milligrams as in our case. Conversely, when expressing the same His-tagged variant in E. coli, 24-28 mg protein/L culture were assessed, which is a sizeable amount (ref. 20).

We judge it more appropriate to comment on these issues in the Section Discussion (lines 312-320) rather than in the Introduction.

-- Introduction: L70: The authors may want to explain how their purification process will impact the stability of the vault nanoparticle based on various storage conditions.

Plausibly, our purification procedure should ensure a very good preservation of protein stability, due to the fact that vault manipulation is reduced to a minimum, in particular if compared with those encompassing several centrifugation steps. Furthermore, vault precipitation from freshly SEC-eluted preparations was barely detectable (as mentioned in lines 158-161), suggestive of a fully (or largely) native conformation. On the other hand, we expect that, at least in principle, if one purifies the vault nanoparticle by a procedure whatsoever and immediately thereafter makes sure that it has retained its native conformation (as we did), its stability should not depend hereafter on the procedure adopted. On the other hand, as far as we know no data are available in literature on vault’s stability under the same conditions we report in Table 1.Comments on these topics are also added in the Discussion (lines 363-370).

-- Results: Sec. 2.1: The title should use the same style of fonts. Please see “K. phaffii”.

Corrected (line 85).

-- Figure 1: Is it possible to add the scale, for example “kDa” in the vertical axis?

In Figure 1, native electropherograms are shown. Thus, the migration is a complex function of both size and charge, which makes it impossible to relate migration to molecular weight. This why no scale was inserted.

-- Figure 3: Is it possible to move the “A1” and “A2” label out of the TEM image in order to make the presentation style consistent? Moreover, I cannot see Figure S1 in the manuscript.

We have moved A1 and A2. Furthermore, Figure S1 is presented in the file “Supplementary information” and is mentioned in the main text (line 143 and legend to Figure 3).

-- Table 1: Data are missing in the second column for the 2nd to 5th row, though I understand the reason why the data of 7 weeks is missing for the 1st row.

Measurements reported in Table 1 have been designed to spot out the condition(s) capable of preserving at best vault’s stability during storage. When stored at 4 °C, already after one day the nanoparticle underwent extensive precipitation and even more after one week, which made it unnecessary to further monitor its behavior. This is why we did not perform measurements at longer times. Conversely, we explored vault stability under frozen conditions. In this case, it made sense to assess not only the effect of freezing in itself, which was done after just a one-day storage, but also the one of a (relatively) long-term storage, which was done after 7 weeks.

Figure 4: I cannot see Figure S2 in the caption of Figure 4 in the manuscript.

Figure S2 is mentioned both in the text (line 208) and in the caption to Figure 4. Obviously, the figure is in “Supplementary information”.

Materials and Methods, L433: “4. Transmission …” should read “4.7. Transmission …”.

Yes, thank you. We have corrected (line 457).

Round 2

Reviewer 3 Report

The authors have addressed most of my concerns. The quality and presentation of this manuscript are improved.